# Accurate Maps of Reef-Scale Bathymetry with Synchronized Underwater Cameras and GNSS

Gerald A. Hatcher [1,*] , Jonathan A. Warrick [1], Christine J. Kranenburg [2] and Andrew C. Ritchie [1]

1 U.S. Geological Survey, Pacific Coastal and Marine Science Center, Santa Cruz, CA 95060, USA; jwarrick@usgs.gov (J.A.W.); aritchie@usgs.gov (A.C.R.)
2 U.S. Geological Survey, St. Petersburg Coastal and Marine Science Center, St. Petersburg, FL 33701, USA; ckranenburg@usgs.gov
* Correspondence: ghatcher@usgs.gov

**Abstract:** We investigate the utility of towed underwater camera systems with tightly coupled Global Navigation Satellite System (GNSS) positions to provide reef-scale bathymetric models with millimeter to centimeter resolutions and accuracies with Structure-from-Motion (SfM) photogrammetry. Successful development of these techniques would allow for detailed assessments of benthic conditions, including the accretion and erosion of reefs and adjacent sediment deposits, without the need for ground control points. We use a multi-camera system towed by a small vessel to map over 70,000 m² of complex shallow (2–8 m water depth) bedrock reef, boulder fields, and fine (sand and gravel) sediments of Lake Tahoe, California. We find that multiple synchronized cameras increase overall mapping coverage and allow for wider survey line spacing. The accuracy of the techniques was sub-millimeter for local length measurements less than a meter, and the bathymetric reproducibility was found to scale with the accuracy of GNSS (3–5 cm), although this could be improved to sub-centimeter with the inclusion of one or more co-registered, but unsurveyed, control points. For future applications, we provide guidance on conducting field operations, correcting underwater image color, and optimizing the SfM workflows. We conclude that a GNSS-coupled underwater camera array is a promising technique to map shallow reefs at high accuracy and resolution without ground control.

**Keywords:** underwater photogrammetry; Structure-from-Motion; underwater mapping; orthomosaic; digital surface model

## 1. Introduction

Camera-based mapping has become an important monitoring tool for shallow reefs owing to technical developments of digital camera systems and Structure-from-Motion (SfM) photogrammetry [1–12]. There are two general image collection strategies for mapping shallow reefs: (i) aerial imaging from above the water surface with a drone or other airborne platform and (ii) underwater imaging from either a swimmer or a piloted vessel. Underwater camera applications eliminate the need to correct for the complex distortion of light at the air–water interface, which can be accomplished, albeit at high computational and data collection costs [5,11,13]. A common challenge of underwater applications, however, is the inability to precisely calculate the locations of the camera, which makes registration of the study site limited to field-placed and difficult-to-survey ground control points (GCPs) and/or scale bars [2,3,10,12,14–16]. As a result, these types of SfM products often have locally based coordinate systems, which are effective for developing accurate measurements of reef geometry and complexity, but require surface fitting approximations, long-term mounted GCPs, or other techniques to co-register or geo-register survey data. As such, it can be difficult and labor intensive to detect benthic changes that are comparable to the mm-to-cm scale changes that occur with coral growth and erosion. The integration of highly accurate geospatial positions from syncing underwater cameras with

Global Navigation Satellite Systems (GNSSs) has allowed for accurate spatial scaling and georeferencing in SfM-derived underwater maps, even without GCPs [12,14]. However, these recent advancements in underwater SfM georeferencing stem from surveys of small patches of reefs (O(100 m$^2$)) [12,14], and it is not known whether these techniques can be scaled up to reef-scale features (O(100,000 m$^2$)).

In this paper, we examine the applicability of a towed imaging system to accurately map reef-scale benthic features using SfM photogrammetry without GCPs. We use a multi-camera towed system that is tightly synced to a survey-grade GNSS, called the SfM Quantitative Underwater Imaging Device with Five Cameras (SQUID-5; Figure 1), that was developed, built, and tested by the U.S. Geological Survey [12]. The SQUID-5 system was designed to collect imagery within several meters of the seafloor or lakebed, which allows natural colors to be well preserved and the imagery ground sample distance to be sub-millimeter to several millimeters. Initial field testing of the SQUID-5 system occurred in shallow reefs of the Florida Keys where short, repeated, overlapping survey lines were used in small (~100 m$^2$) areas over targets of interest. For this to be a useful tool for reef research and monitoring, imaging systems need to be able to collect contiguous data over areas at least as large as those collected using traditional scuba or snorkel surveying techniques, and preferably much larger, i.e., 10 s to 100 s of meters in length and 1000 s to 10 s of thousands of square meters. In this study, we use SQUID-5 to map almost 100,000 m$^2$ of a shallow bedrock reef of Lake Tahoe, California, over two days of field operations. This site was chosen because of the lake's clear water, the complex erosional features of the mudstone reef that are found in approximately two to seven meters of water depth, and the limited ability to travel to other sites during the COVID-19 pandemic (Figure 2).

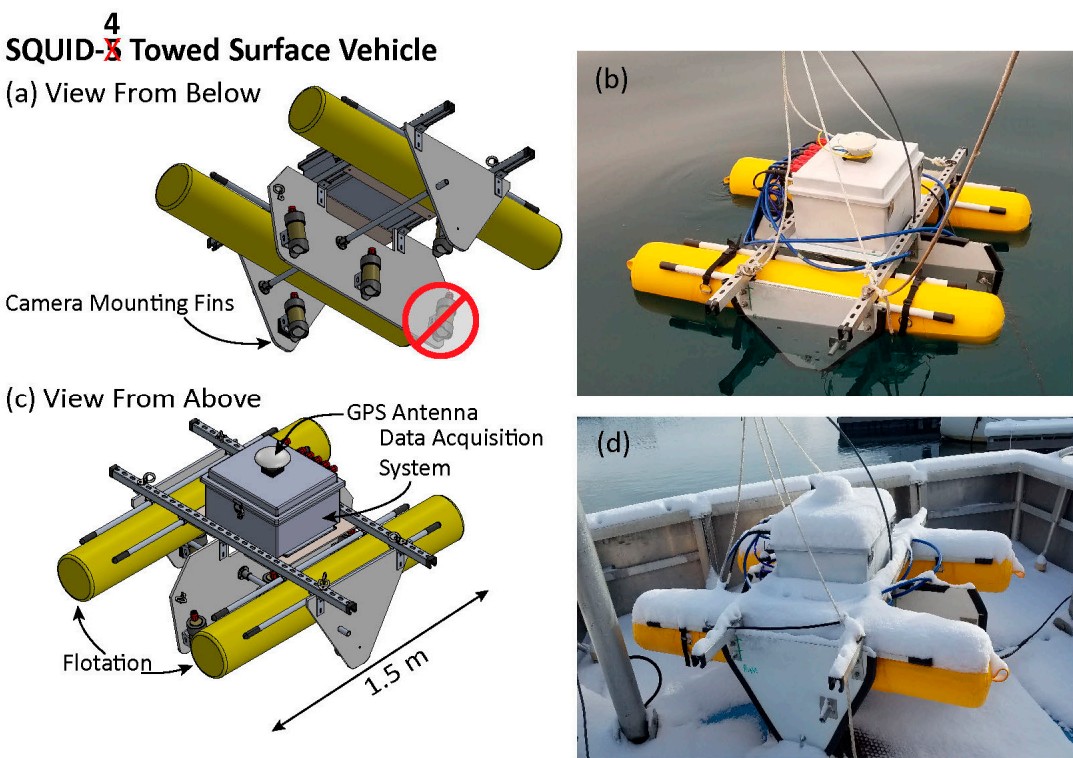

**Figure 1.** Engineering drawings and photography of the SQUID-5 system used for benthic photogrammetric mapping. During the operations at Lake Tahoe for this study, one of the five cameras did not function, as suggested by the notes on the diagrams, essentially making the system a four-camera system (or "SQUID-4") for the survey. Panels (**a**,**c**) illustrate the relative layout of major system components of the towed surface vehicle. Panel (**b**) shows the towed surface vehicle as it floats on the water surface. Panel (**d**) shows the towed surface vehicle covered with snow while sitting on the support vessel before deployment.

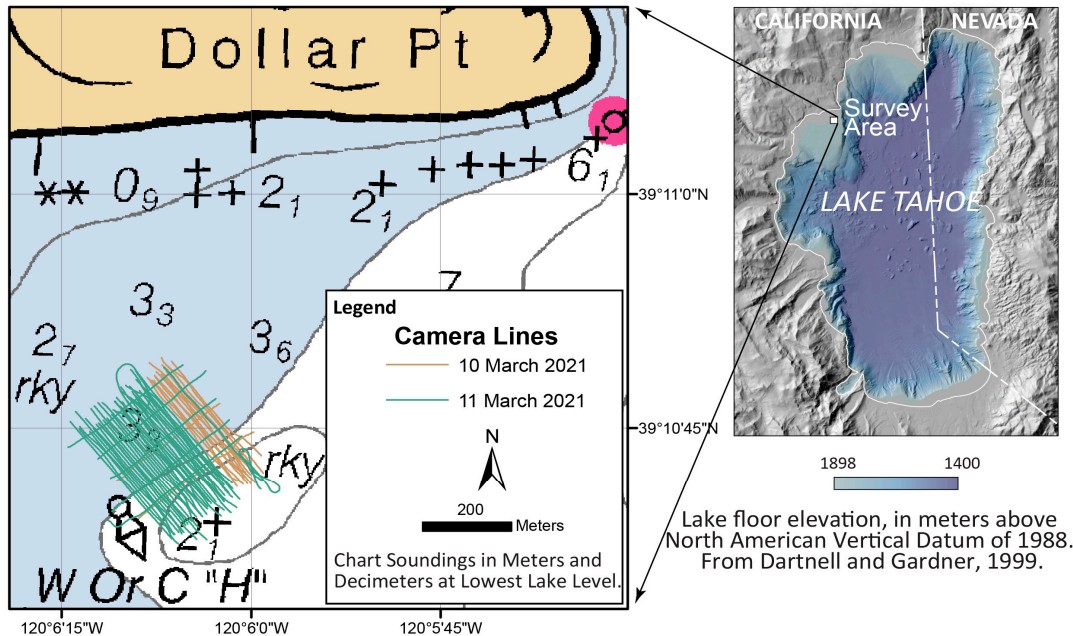

**Figure 2.** Lake Tahoe, California, survey area and track-line map. Survey track lines are shown with orange (day one) and green (day two) lines over the NOAA nautical chart for Lake Tahoe, California.

The goal of this work was to develop an example data set that could be used to optimize field operating procedures, refine data processing methods, and assess uncertainties in the products. Three main challenges were expected in scaling up the technique. First, it was unknown if the rate of image data collection required for overlapping bottom coverage could be reliably managed over a sustained period. Second, it was unknown whether it was reasonable to accurately maneuver a small research vessel while towing the camera system to generate complete survey area coverage. Third, it was unknown whether data processing techniques could be applied to the several tens-of-thousands of overlapping images that would be collected, including overnight processing while in the field to provide daily updates about survey collection coverage. Below, we describe the collection protocols, outline post-processing guidelines, discuss data products, and evaluate the effects of color correction and different camera arrangements.

## 2. Methodology

### 2.1. Instrumentation

Detailed descriptions of the SQUID-5 system are provided in Hatcher et al. [12], including engineering details found in the Supplemental Materials of that publication. Although we will not repeat these details here, we do provide a general description and report updates and modifications of the system. In general, SQUID-5 utilizes a rigid-framed vessel that is towed by a support boat providing power and remote-system controls (Figure 1). Imagery is obtained from five synchronous 5.0 MP Teledyne FLIR™ visible color spectrum machine vision cameras, with the Sony™ IMX 264 CMOS sensor. Higher MP cameras were not required due to the limited distance from the seabed to the cameras. The lower-pixel-count cameras allowed for a faster sustained capture rate and reduced the overall data volume, while still providing sub-mm ground sample distances.

The small size of the cameras (approximately a 30 mm cube) allows for custom underwater housings, with hemispheric glass domes to minimize optical distortions. Positions of the cameras during operations are determined with the combination of survey-grade GNSS positions provided by a Trimble R7 receiver and Zephyr-2 antenna mounted onto the SQUID-5 system, and measured offsets between the antenna and the camera nodal points. Cameras are operated to collect synchronous imagery at 1 Hz. Tight

synchronization between the cameras and GNSS resulted in positional uncertainties from temporal latencies that were more than two orders of magnitude less than the GNSS horizontal uncertainty of approximately 1.7 cm.

A primary update to the SQUID-5 system for this work was a comprehensive control and acquisition program that made field collections more efficient [17]. The configuration of the navigation display was also upgraded to include a live stream of GNSS data to help with the difficult task of accurately towing the SQUID-5 sled over small target areas in real time. Additionally, while processing data from initial fieldwork in 2019, it was discovered that the center camera was often redundant with the other cameras and, in some instances, would introduce noise to SfM products from oversampling. Therefore, during the 2021 field operations, we investigated whether changing the center camera lens to a longer focal length (from 6 to 8 mm) would reduce redundancy and improve the data quality and resolution.

Operating the SQUID-5 system somewhat continuously over large areas provided the challenge of adequately collecting and managing the large volume of raw image data expected each survey day. For example, a single uncompressed RGB image requires approximately 15 MB of data storage, so collecting imagery at 1 Hz simultaneously from five cameras generates approximately 75 MB/s or 270 GB/h. Therefore, storage devices needed to be extremely fast to facilitate the terabyte scale of data collection and backup required each survey day. To facilitate this, a data acquisition computer mounted directly on the SQUID-5 tow vehicle stored the image data locally to an internal striped RAID 0 array comprised of four high-speed solid state hard drives. Although RAID 0 has the fastest throughput speed of all the present RAID configurations, it also has the highest probability of failure since the data are striped across all disks in the array, so failure of any one disk destroys the entire file structure. To reduce the risk of data loss, we included an additional backup storage system incorporating four high-speed solid-state hard drives configured as RAID 5 which is slightly slower but able to recover from a single disk failure. The backup storage system was co-located with the shipboard equipment in the protected lab space, and this network-attached RAID 5 storage system was regularly updated with backup copies of new data while the survey was underway. This resulted in the coincidental generation of backup data copies and new data acquisition. A third copy of the daily data was created at the end of each day on a single solid-state drive connected directly to the shipboard RAID 5 array using an eSATA connection. The goal was to process the third copy onshore each evening and generate initial provisional SfM alignment products to guide the survey by identifying data gaps or problem areas needing to be re-mapped.

*2.2. Study Site*

Owing to travel limitations imposed by the COVID-19 pandemic and the desire to use SQUID-5 in future coral reef mapping efforts, we attempted to find a location that closely resembled coral reef condtions. Based on experience gained mapping the Florida Keys, USA, in 2019 [12], desirable study site characteristics included (i) sufficient water clarity, such that the bottom can be imaged from just below the surface using natural sunlight in depths as great as 8 meters; (ii) a navigable area of at least 200 m in length and width, with the majority of its water between 3 and 8 m of depth; (iii) stationary bottom features with adequate relief and texture to be reasonable proxies for corals and reef features; and (iv) accessibility for a support vessel that has weather-protected space large enough to accommodate two survey personel, a boat captain, acquisition computers, navigation equipment, digital displays, and deck space to launch and recover the SQUID-5 vehicle. The shallow, bedrock reefs of northern Lake Tahoe, USA (Figure 2), fit these requirements, because it is an ancient volcanic lahar formation that has been eroded into complex high-relief features. Additionally, a collaboration with the UC Davis Tahoe Environmental Research Center (TERC) and use of their 12-meter (40-foot) support vessel R/V John Le Conte ensured our ability to capture this area with tow speeds as low as 1.5 m/s.

### 2.3. Field Operations

We conducted two sampling exercises to map the study area with the SQUID-5 system. The first field operation was conducted on 14–18 September 2020, but was hampered by excessive smoke conditions from local wildfires, which compromised our ability to conduct the survey, and GNSS data problems related to an unsuccessful attempt to integrate a low-cost dual-frequency GNSS with our survey-grade GNSS for testing purposes. In the end, a signal splitter used to incorporate both GNSSs resulted in compromised GNSS data that could not be used to generate accurate position data from either receiver. However, the limited data collection that occurred during this operation allowed us to realize that (i) the upgrades to our system, including data backup and data transfer, were successful; (ii) a field data collection and overnight data processing workflows were possible with a land-based processing computer (Figure 3); (iii) the camera settings had optimal ranges of exposure times and sensor gains; and (iv) the 3 m line spacing used for data collection was generally adequate, as shown by provisional SfM alignment tests.

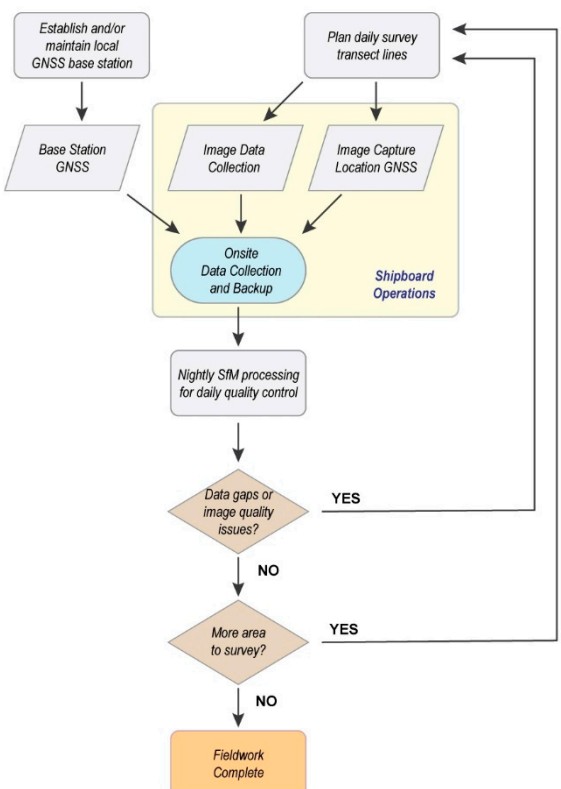

**Figure 3.** Flow diagram for field operations with the SQUID-5 system.

Thus, a second operation was conducted on 8–12 March 2021, which provided the data utilized in the analyses below. Conditions during these operations included periods of significant snowfall (Figure 1d), which reduced visibility, and periods of gusty winds, which caused sustained surface waves as large as 0.5 m. The snow squalls were separated by short periods of sunny, clear blue sky that made the underwater lighting environment highly variable. The survey line layout included a series of 5 m spaced, northwest-to-southeast-trending parallel survey lines and several perpendicular lines across the survey area, which were completed with two days of survey operations (Figure 2).

The cameras used are all the Teledyne FLIR ™ BFS-PGE-50S5C-C, with the Fujinon HF6XA-5M 6 mm fixed focal length lens on the forward and two outward cameras and the Fujinon HF8XA-5M 8 mm fixed focal length lens on the center downward camera. The system was towed at aproximately 1.5 m/s. To adjust for changing light conditons, the preference was to adjust only the sensor gain when possible and change the exposure time

setting only when absolutely necessary for correct exposure. The exposure was adjusted for a slightly dark image to reduce the possibility of sensor saturation and image data loss. Based on results from both field collections, we empirically determined the optimal camera settings for our condions, listed in Table 1.

**Table 1.** Prefered ranges of camera operating parameters settings.

| Camera Teledyne FLIR ™ BFS-PGE-50S5C-C. Tow Speed ~1.5 m/s. | | |
|---|---|---|
| Lens | HF6XA-5M 6 mm | HF8XA-5M 8 mm |
| Exposure (microseconds) | 1000–3134 | 1000–3134 |
| Sensor Gain (dB) | 12–18 | 12–18 |
| f-Stop | 5.6 | 5.0 |
| Collection Rate (Hz) | 1 | 1 |

### 2.4. Field Data Collection

During the March 2021 field operations, a local GNSS base station was temporarily installed at the TERC field station, which is roughly 2 km away from the survey site. The base station enabled location data from SQUID-5 to be corrected using Post Processing Kinematic (PPK) methods. Additionally, a NOAA Continuously Operating Reference Station (CORS) located within 15 km of our survey (P150) area was available for use during post-processing to generate final data products once we returned from the field. The raw GNSS data from the SQUID-5 were minimally processed each day to verify their integrity and to provide assurance that the event marks of image capture locations were properly logged (Figure 3). Additional GNSS processing was conducted after the cruise for SfM data analyses, as described in Section 2.5 below.

In addition to the SQUID-5 system changes described above, we also included a dual-antenna GNSS for the R/V John Le Conte, which provided real-time heading information, in addition to position, which were displayed in the live navigation software and allowed for better navigation. Unfortunately, the rear camera connection of SQUID-5 was found to be faulty immediately before the field operations, which limited our data collection to only the left, right, center, and front cameras (Figure 1a). Otherwise, the survey was conducted as planned, resulting in 88 closely spaced survey lines at approximately 5 m line spacing and over an area approximately 250 meters by 235 meters (Figure 2). The combined line spacing and camera geometry of the SQUID-5 system resulted in approximately 80% overlap and 50% sidelap in the imagery set collected from the multiple cameras, although these values varied spatially owing to the actual line spacing and water depths during the survey. The central camera, which obtained higher-resolution imagery owing to its lens, resulted in approximately 50% overlap and 20% sidelap, independently from the other cameras, suggesting that most of the study area would have been covered with these higher-resolution images. The raw camera imagery and post-processed GNSS position data are published and available, along with extensive metadata, in a USGS Data Release publication [18].

### 2.5. SfM Data Processing

Our SfM data processing techniques follow the general considerations and workflow of Over et al. [19], with modifications for multiple-camera underwater operations, as suggested by Hatcher et al. [12]. The general workflow is represented graphically in Figure 4, and further details are provided below. A key element of this workflow is the development of geospatial positions for the location of the SQUID-5 GNSS antenna for each image capture, because these data are used by the SfM software (Agisoft Metashape version 1.6.4.) with camera-to-GNSS antenna offsets, also referred to as "lever arms," to compute the camera nodal positions for all images. These positions are derived from GNSS time event records recorded during field operations at the precise time of each image collection, and these positions are introduced to the SfM project by reconciling the positions with image file names (Figure 4). The GNSS position data were processed after

the precise ephemeris, and clock data were available using Post Processing Kinematic (PPK) techniques with NovAtel's GrafNav software. GrafNav robustly filters and corrects outliers, making extensive use of Kalman filtering, and resulted in estimated 2-sigma uncertainties of 10 and 15 cm in the horizontal and vertical directions, respectively. As noted below, the reconciled GNSS positions were used as initial locations for each of the cameras and were refined further in the SfM alignment and optimizations, which resulted in mean position uncertainties of approximately 3 cm.

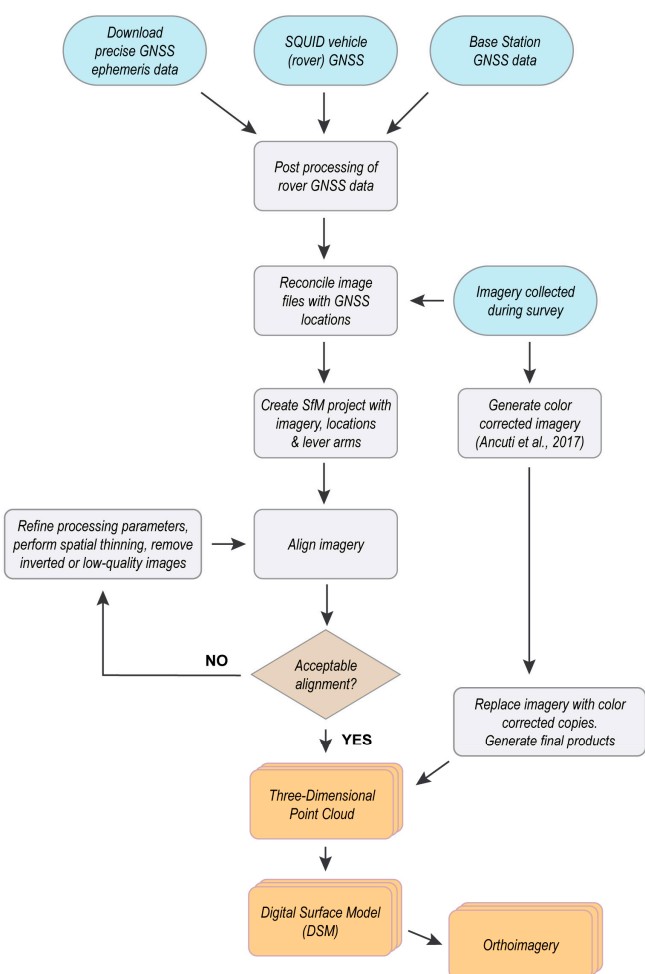

**Figure 4.** Data processing flow diagram for Structure from Motion (SfM) analyses and products of data obtained from the SQUID-5 system [20].

A total of 42,939 images and associated positions were collected within the study area during the March 2021 operations from the four cameras, and these data were used for SfM alignment, optimization, and product generation. Following the workflows of Hatcher et al. [12], imagery were aligned in Metashape using the high setting, a maximum of 60,000 key points per image, unlimited tie points, and preselecting image-to-image matching using the imagery position data (Figure 4). Additionally, we allowed the measured lever arm lengths to be adjusted with a total accuracy of 2.5 cm during the alignment, owing to the potential for physical shifting of the SQUID-5 system during transport and set up. We also assigned the camera positional accuracy to a conservative 2 cm in the horizontal and 6 cm in the vertical. This resulted in over 94.5 million tie points, or an average of ~2200 points per image. Tie points with the greatest uncertainty were removed using Metashape's gradual selection tools, which were set to recommended thresholds for underwater collections: reconstruction uncertainty (RU) of 20, projection accuracy (PA) of 8, and reprojection error (RE) of 0.4 [12]. Camera optimizations were completed twice, once

after RU and PA were applied and once after RE, always using standard lens distortion parameters (f, cx, cy, k1, k2, k3, p1, p2). This reduced the number of tie points to slightly over 62.5 million and resulted in low camera position errors (x = 0.65 cm, y = 0.94 cm, z = 3.02 cm, total = 3.23 cm) and low camera lens residual errors (all residuals throughout each 2D model were less than 0.25 pixels). We note that the alignment process may identify poor-quality imagery or position data, or the gradual selection process may remove too many tie points for an adequate alignment, and under these conditions, users may revise PU, PA, or RE settings; thin imagery to a desired level of overlap in the along- or across-transect direction ("spatial thinning"); or remove erroneous imagery or position data, before re-aligning the data (Figure 4) [12,19]. Our data collection did not require any of these corrections.

Because of the strong color modifications caused by light absorption and scattering in underwater imagery, a color correction process was conducted on the raw images before generating point cloud and orthoimage products (Figure 4). The color correction technique was a twofold process. First, images were corrected for the high absorption (and low color values) in the red band using the color balancing techniques of Ancuti et al. [20]. For this, the red channel was modified using a color compensation equation (Equation (4) of Ancuti et al. [20]) that uses both image-wide and pixel-by-pixel comparisons of red brightness with respect to green brightness. After compensation, the images were white balanced using the "grayworld" assumption [20], which ensures that the three-color-band histograms are centered on the mean brightness value of the image. The remaining techniques of Ancuti et al. [20], which include sharpening and multi-product fusion, were not employed. The resulting images utilized only about a quarter to a half of the complete 0–255 dynamic range of the three-color bands. Thus, the brightness values of each band were stretched linearly over the complete range, while allowing the brightest and darkest 0.05% of the original image pixels (i.e., 2506 of the 5.013 million pixels) to be excluded from the histogram stretch. This final element was included to ensure that light or dark spots in the photos, which often occurred from water column particles or image noise, did not exert undue control on the final brightness values. Examples of the color correction output are included in Figure 5.

Metashape was then used to generate a 3D point cloud using the high-resolution and moderate filtering settings [12,19]. This resulted in a total of 3.63 billion points for the 77,400 m$^2$ study area, and each point included a RGB color value derived from the color-corrected photos and a confidence estimate, which was equivalent to the number of photos that were included to generate each point. Examples of the RGB colors and confidence values of small sections of the point clouds are shown in Figure 6. Low point confidence was found in two general areas: (i) near vertical and overhanging rock faces, where few images were able to resolve the feature (Figure 6, lefthand column), and (ii) areas where poor alignment made short (several meters in length) linear offsets at the outermost edge of image projections (Figure 6, righthand column). The occurrence of (i) was persistent through the study area, whereas less than a dozen examples of (ii) could be found. Both areas of low confidence were represented by noisy output in the point clouds that did not mimic lakebed morphology (Figure 6). Thus, we classified the point clouds based on confidence values, and defined "noise" points to be products of only one image and "lakebed" points to be products of two or more images (Figure 6e,f). This resulted in about 425 million, or ~11.7% of the points, classified as noise. The resulting lakebed point cloud had an average point cloud density of 4.1 points/cm$^2$.

The lakebed point cloud was used to generate digital surface models (DSMs) of the study area, using Metashape with interpolation disabled [19]. Using the high-resolution lakebed point clouds, Metashape generated a DSM with 5.3 mm × 5.3 mm pixel resolution. This DSM was output at 25 mm × 25 mm pixel resolution to produce a manageable file size (311 MB compressed), although different resolution DSMs can be made with the raw point cloud data.

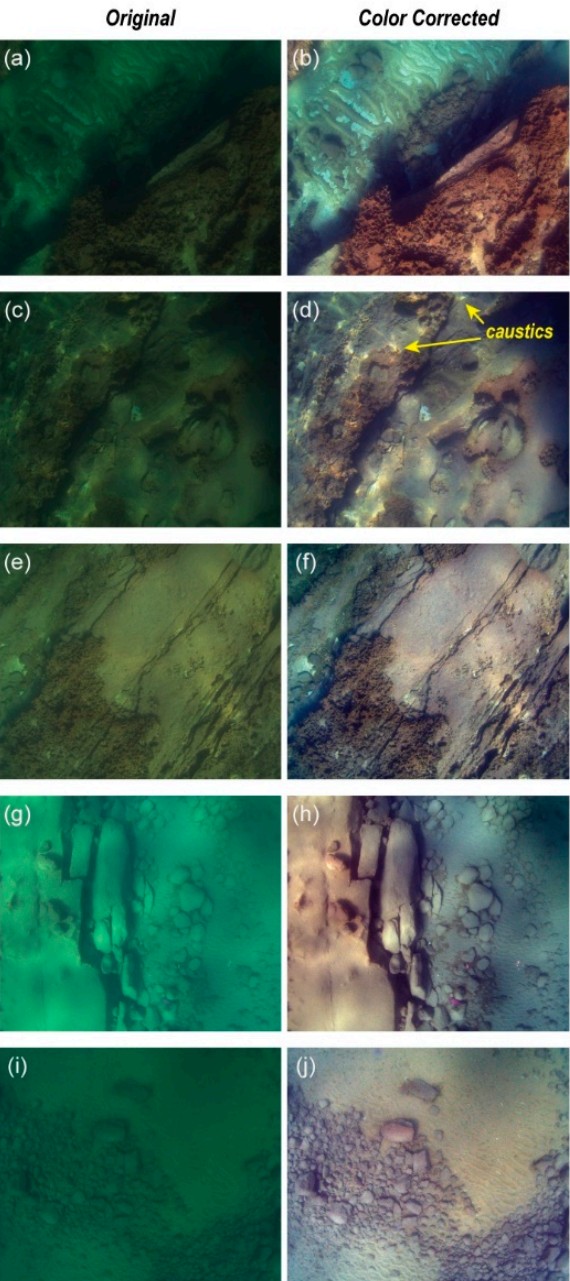

**Figure 5.** Examples of raw original imagery of the lakebed of the study area (**a,c,e,g,i**) and color-corrected versions of the imagery (**b,d,f,h,j**) using the techniques described in the text.

Orthoimages were then made with Metashape, using the color-corrected imagery projected onto the DSM (Figure 4). Two orthoimage creation techniques were used and are compared below: "orthomosaic" and "ortho-average". The orthomosaic technique blends low-frequency components of overlapping images using a weighted-average algorithm and uses the high-frequency component from the single image most normal to the viewpoint. These imagery data are projected over polygonal areas of the study that are separated by automatically generated seamlines. We did not employ any seam refinement in the orthomosaic technique. The ortho-average technique computes an RGB color for each pixel of the orthoimage from the average RGB color of all photos contributing to it. For both methods, we did not use hole filling options over the gaps in the DSM. The resulting orthoimages were generated at a 2.6 mm × 2.6 mm pixel resolution, and we output these products at a 5 mm × 5 mm pixel resolution to produce manageable file sizes.

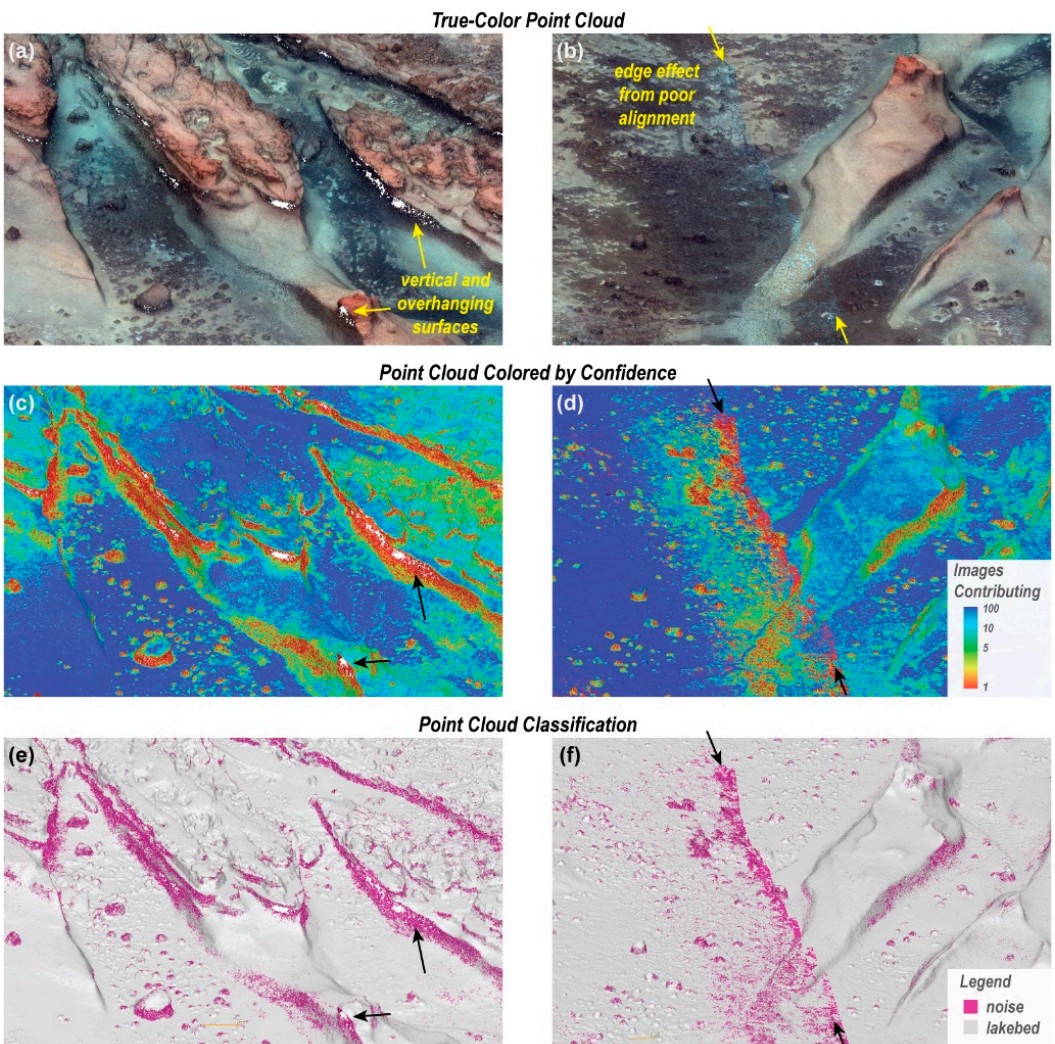

**Figure 6.** Oblique perspectives of example Structure from Motion (SfM) point cloud outputs for the Lake Tahoe study area highlighting the (**a**,**b**) true-color point clouds, (**c**,**d**) Metashape confidence values of each point, and (**e**,**f**) our classification of noise points based on the confidence values. Two examples of noisy data are shown and highlighted with arrows: vertical and overhanging surfaces, where features are commonly resolved by only a single image, and linear offsets in the SfM output associated with the edges of images that were not well aligned. Both point types are classified as noise and excluded from DSM products.

The resulting 3D point clouds, DSM, and orthoimagery products are published and available in a USGS Data Release publication [21], and digital mapping products of these data are provided in a USGS Scientific Investigations Map [22]. Lake elevations are reported with respect to the North American Vertical Datum of 1988 (NAVD88), rather than water depths, which fluctuate by several meters over annual and interannual scales. For conversion purposes, however, the Lake Tahoe water level during our field operations averaged 1898.44 m NAVD88, as measured by the USGS station 10337000, which is named Lake Tahoe at Tahoe City (data available from U.S. Geological Survey, 2022 [23]).

### 2.6. Additional Tests

In addition to the general data processing described above, we also conducted several tests to evaluate the data quality of the products and the effects of data collection and data processing methods. First, we utilized the machined aluminum and painted "Picasso Plate", described fully in Hatcher et al. [12], to provide assessments of horizontal and

vertical distance measurements on the order of decimeters in the SfM products. The Picasso Plate has four targets, three of which are on corners of the plate and one in the center on a 10 cm high platform, and the relative positions of which were machined to sub-millimeter accuracies [12]. This plate was placed twice in the study area, once on each survey date, so that it would be fully imaged in at least two survey transects. After SfM alignment, digital markers were placed on the locations of the four machined target locations, and distances between the corner markers and the height of the center marker were calculated from the positions derived from the SfM data. Comparisons between actual distances and SfM-derived distances were tabulated for assessments of local-scale distance measurement accuracies in the SfM products.

We also ran tests to evaluate the utility of multi-camera setups by rerunning the complete SfM data processing workflows with only one, two, and three cameras. For these SfM analyses, we used the middle camera for the one-camera processing, the side-looking cameras for the two-camera processing, and two different setups for three cameras: a "linear" compilation of the two side-looking cameras and the middle camera and an "outer" compilation of the two side-looking cameras and the forward-looking camera. To compare these outputs, we evaluated the total coverage as computed by the number of 25 mm × 25 mm DSM pixels produced by each setup and evaluated vertical differences between the DSMs.

Lastly, we examined the repeatability of the SfM results by comparing an area of survey overlap captured during both the first and second days of operations. This area of overlap was relatively small (273 m$^2$, or ~0.35% of the surveyed study area) because it was not a survey goal to include overlap. To make this comparison, the SfM workflow was conducted independently on the complete set of data from the two survey days. Then, DSM data were clipped out of each resulting product for only the area of overlap where both survey days had both continuous data and survey lines bounding each side of the overlap area. Differences between these independent SfM outputs were assessed by computing differences of the DSMs to evaluate vertical offsets, and by marking 8 recognizable features on flat portions of the lakebed, such as bedding or other coloration, to compute horizontal and vertical position differences for each point.

## 3. Results

### 3.1. SfM Mapping

During the two-day cruise, we were able to map an area approximately 240 m × 300 m, with water depths ranging from 2 to 8 m. The total footprint of the mapped area was 77,400 m$^2$, and within this area, 74,900 m$^2$, or 96.8%, had data returns in the 25 mm × 25 mm DSM (Figure 7). Within the broader survey area, the bathymetric data revealed two northwest-to-southeast-trending shallow ridges separated by a deeper trough. Along the northeastern boundary of the study area, there was a steep, ~3 m vertical face with several rockfall blocks along the base of the slope (Figure 7a). Over smaller scales, the lakebed included additional complexity from ridges, boulders, meter-scale round pits, sandy areas with rippled bedforms, and sedimentary bedding (Figure 7b).

Gaps in data output were largely caused by incomplete survey coverage of the SQUID-5 system over the study area, which is shown by several narrow (several meters wide, tens of meters long) along-track holes in the data (Figure 7a). As noted in Section 3.3.2 below, these data gaps were used to assess the operational line spacing that was achieved during the surveys. Other data gaps included small (centimeter-to-decimeter scale) regions around vertical and overhanging lakebed features that were not captured by imagery data. These resulted from the narrow opportunity for imaging these high-relief surfaces with a towed camera sled at the water surface and also from the noise reduction technique that preferentially excluded points from the DSM along these vertical to overhanging faces (cf. Figure 6).

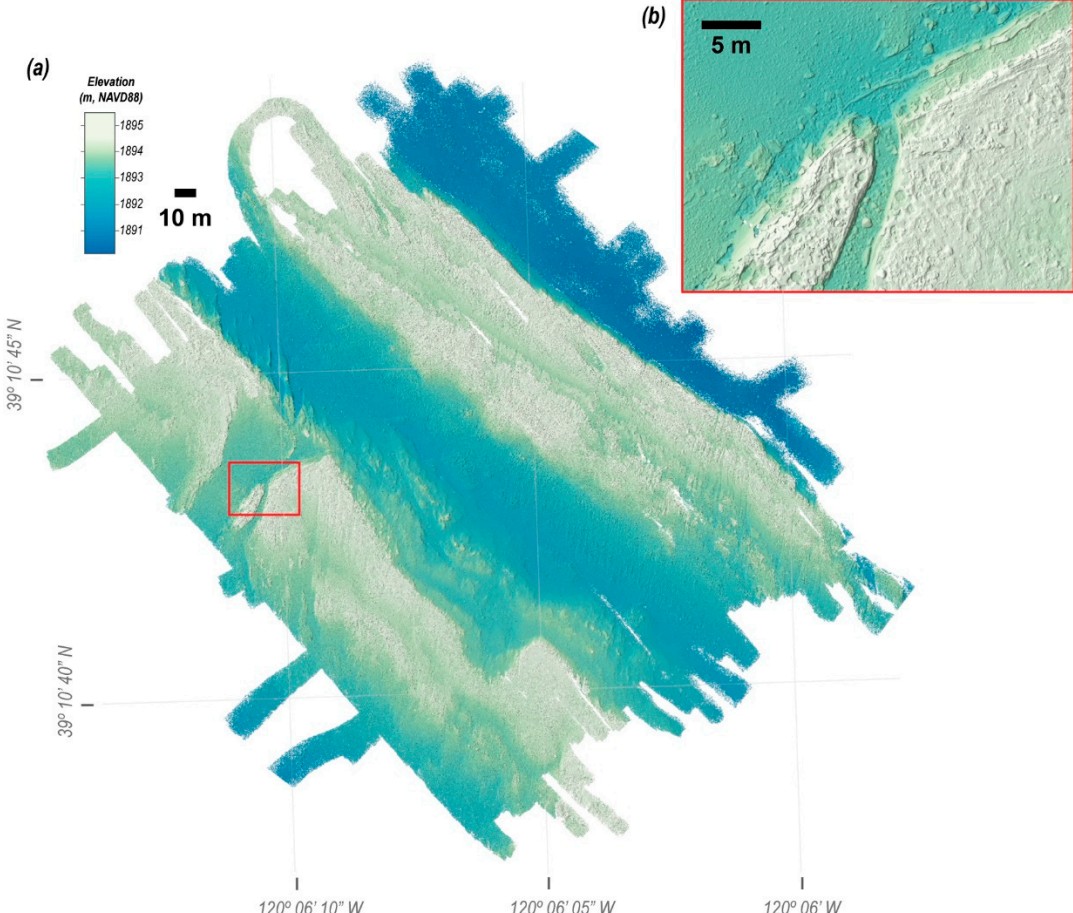

**Figure 7.** (**a**) The Digital Surface Model (DSM) for the complete study area derived during two days of surveying. The mean lake water level during the survey was 1898.44 m NAVD88. (**b**) Inset of a portion of the study area showing further details of the bathymetric characteristics.

The orthoimage products provide sub-centimeter depictions of the lakebed. Orthoimagery of the entire study area is best viewed digitally, because it is impossible to show the details of these data in this print format, so we direct readers to these products in the data release publication [21] and the USGS Scientific Investigations Map [22]. Here, we provide close-up perspectives of these data to highlight differences in the output of the two techniques and to provide descriptions of the study site lakebed morphology. Several differences are observed in the orthomosaic and the ortho-average image products (Figure 8). The orthomosaic has a larger range of color output, whereas the compilation of multiple RGB values into the ortho-average results in a duller, grayer product (Figure 8). However, because the orthomosaic is derived from clippings of individual images, characteristics of these input images are preserved in the orthoimage. For example, there are both areas of highly resolved, clear imagery and blurry imagery in the orthomosaic (Figure 8a), whereas the ortho-average has a more uniform clarity throughout (Figure 8b). Additionally, the orthomosaic preserves patterns of the underwater light caustics, which are areas in which sunlight is focused onto the lakebed (Figure 8a). The ortho-average image does not show caustics, because these features are averaged out, with the inclusion of several to over a dozen images per pixel (Figure 8b). This is important because caustics commonly cause strong gradients in the orthomosaic brightness, and these highly contrasting areas can be difficult to characterize by users.

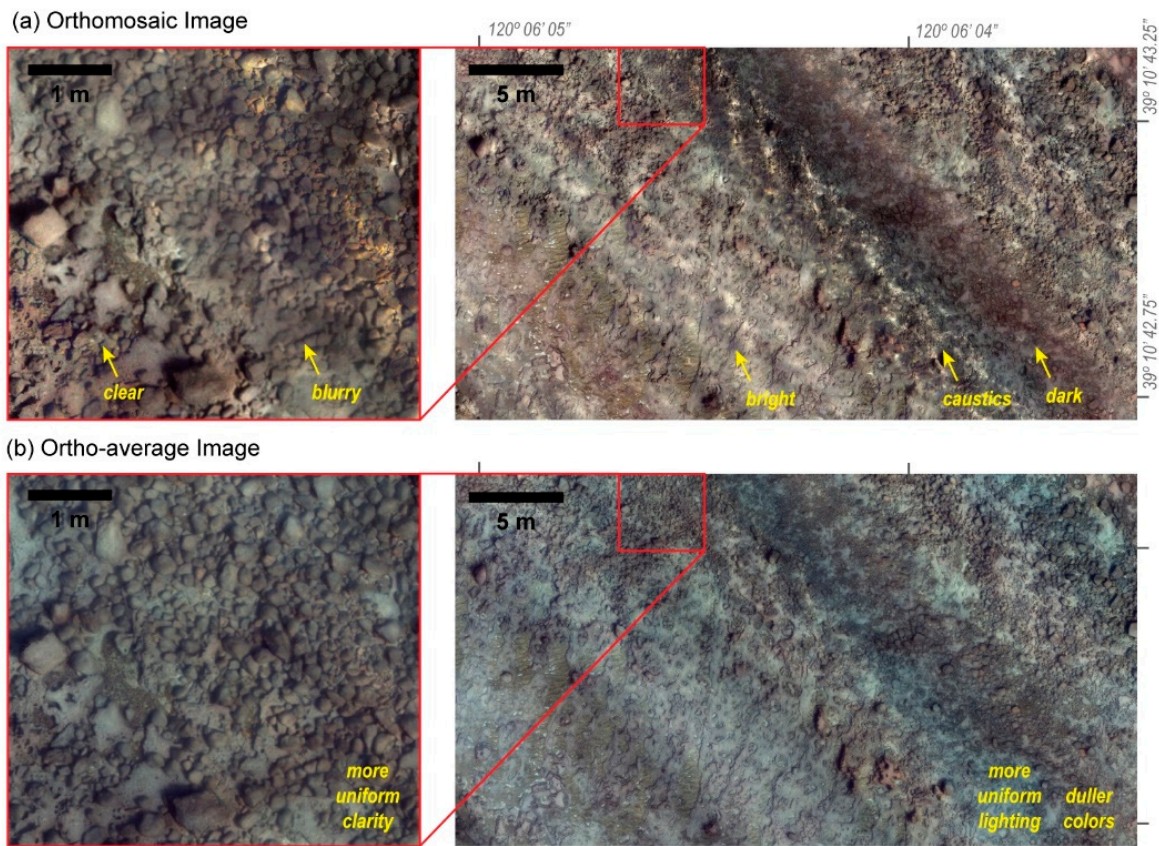

**Figure 8.** Example orthoimagery data from the study area from the two different techniques: (**a**) orthomosaic and (**b**) ortho-average. Differences in these products are highlighted with text and arrows. For a complete study area map of the orthoimagery products, see the USGS Scientific Investigations Map 3501 [22].

Combined, the DSM and ortho-average data provide detailed views of the geomorphic features of the study area (Figure 9). Lakebed characteristics include a diverse set of bedrock exposures, including hollows, smooth exposures, elevated platforms, fissures and cracks, and a broad distribution of sediment, ranging from sand to coarse sediment (gravel, cobbles, and boulders, Figure 9a). Additionally, the data provide evidence of geomorphic processes, including rockfalls along steep vertical faces (Figure 9e) and sediment transport from rippled bedforms (Figure 9f). The bathymetric data about these features are generally continuous, except adjacent to the steepest slopes and within complex areas—as shown with white pixels in Figure 9c–f—where the cameras could not acquire adequate imagery to resolve these features.

### 3.2. Accuracy Assessments

The first accuracy assessment technique utilized the machined Picasso Plate [12] to provide comparisons of sub-meter-scale measurements of distance. The plate was placed twice within the survey area, once on each survey day, and comparisons of the actual machined distances of the plate and the SfM-derived measurements of the plate are provided in Table 2. All horizontal measurements were achieved with sub-millimeter errors, and the mean and standard deviation of these errors were $-0.0002 \pm 0.0003$ m, which are equivalent to $-0.04 \pm 0.05\%$ of the actual lengths. The vertical measurements were not as accurate as the horizontal measurements, and these errors were found to be millimeter-scale, or equivalent to $-0.50 \pm 1.41\%$ of the actual vertical distance (Table 2). This suggests that the SfM techniques without ground control are accurate to at least millimeter scales over

these local, decimeter-scale distances, and that horizontal measurements were generally several times more accurate than vertical measurements.

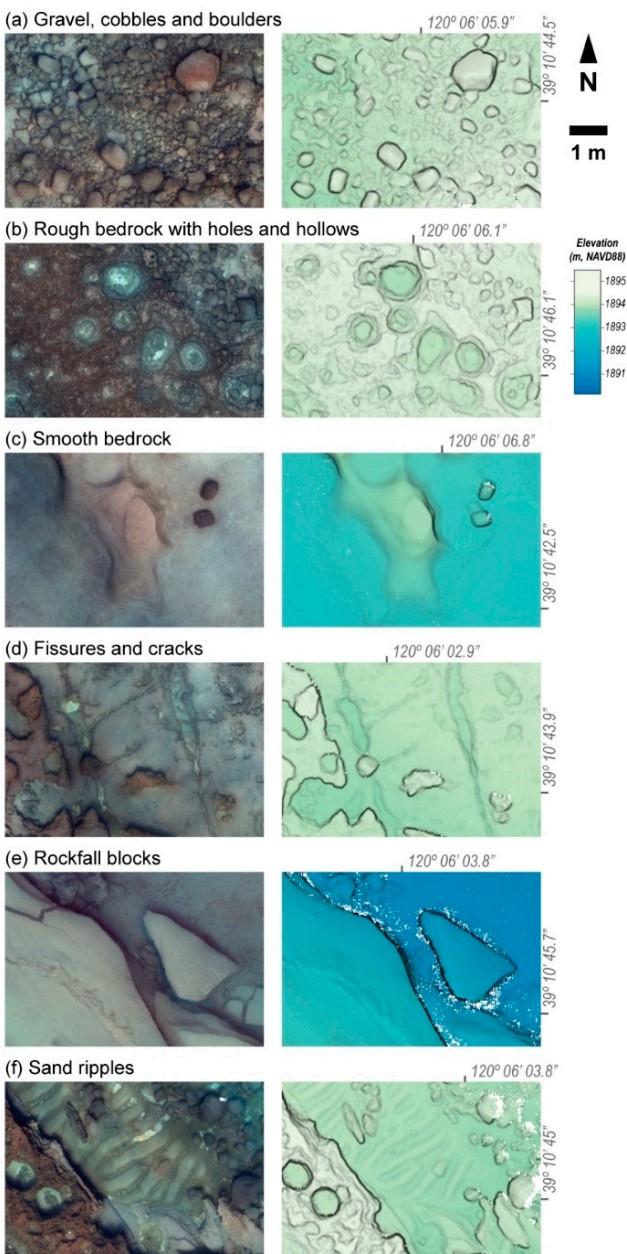

**Figure 9.** Example ortho-average (lefthand column) and Digital Surface Model (DSM) output (right-hand column) of the diversity of geomorphic features within the study area. Data gaps in the DSM are absent in (**a**,**b**) and shown as white pixels in (**c**–**f**).

In addition to the local accuracy assessments, we utilized an area of survey overlap to test reproducibility from independent SfM analyses. Bathymetric maps of this comparative area for both survey days are provided in Figure 10, and this area includes a channel feature, numerous cobbles, and a combination of rough and flat rocky areas. The vertical difference between these two independent mapping products shows that the greatest differences occur along steep slopes and that positive changes are consistently along southwest-aspect slopes, whereas negative changes are consistently along northeast-aspect slopes (Figure 10c). However, the overall vertical change between these DSMs was negligible, as the mean and standard deviation of the differences was $0.004 \pm 0.018$ m. Thus, these patterns of change

suggest that there was a southwest-to-northeast-oriented horizontal offset between the two DSMs.

**Table 2.** Measurements of two placements of the Picasso Plate for assessments of local distance measurements: actual measurements from machined target center points, measured distances from Metashape markers placed on targets in Structure from Motion (SfM) output.

| Plate Measurement | Actual Distance (m) | Measured Distance (m) | Error (m) | Error (%) |
|---|---|---|---|---|
| Short Axis, Day 1 | 0.4000 | 0.3997 | −0.0003 | −0.08% |
| Short Axis, Day 2 | 0.4000 | 0.4000 | 0.0000 | 0.00% |
| Long Axis, Day 1 | 0.6000 | 0.5996 | −0.0004 | −0.07% |
| Long Axis, Day 2 | 0.6000 | 0.6001 | 0.0001 | 0.02% |
| Diagonal, Day 1 | 0.7211 | 0.7204 | −0.0007 | −0.10% |
| Diagonal, Day 2 | 0.7211 | 0.7211 | 0.0000 | 0.00% |
| Vertical Platform, Day 1 | 0.1003 | 0.0988 | −0.0015 | −1.50% |
| Vertical Platform, Day2 | 0.1003 | 0.1008 | 0.0005 | 0.50% |
| Horizontal Mean ± St. Dev. | | | −0.0002 ± 0.0003 | −0.04 ± 0.05% |
| Vertical Mean ± St. Dev. | | | −0.0005 ± 0.0014 | −0.50 ± 1.41% |

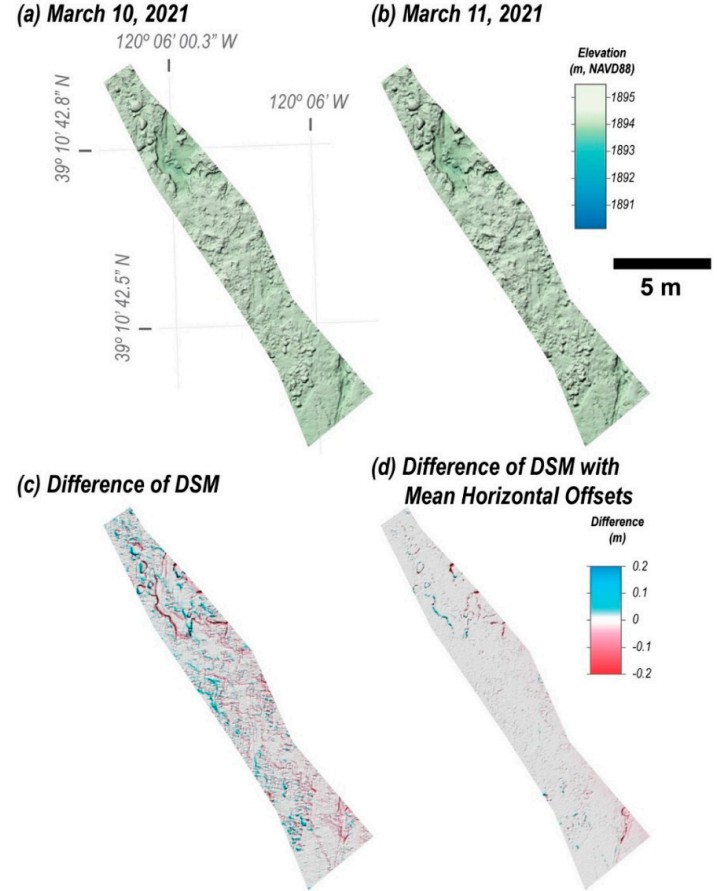

**Figure 10.** Comparison of the area of overlap between day 1 and day 2 of the field operations. Digital Surface Model (DSM) products shown in (**a**,**b**) were generated with independent Structure from Motion (SfM) data processing, thereby resulting in independent results. (**c**) The differences of the two DSMs. (**d**) The differences in the DSMs after the horizontal offsets between the two data sets were applied (see text).

This horizontal offset in the two DSMs is confirmed with measurements of positional differences from eight markers placed on distinctive geographical features in each mapping

area (Table 3). Consistent offsets were found for both eastings and northings, resulting in a total horizontal offset of 0.047 ± 0.001 m, and the difference in the elevation was found to be insignificant (0.001 ± 0.010 m; Table 3). If the mean horizontal offsets are applied to the DSM products, much of the vertical change is eliminated, and the variance in the differences between the DSMs is reduced from 0.018 to 0.007 m (Figure 10c,d). The vertical offsets remaining after this horizontal shift largely occur along steep slopes, which may be related to poor data quality along these difficult-to-photograph features, as noted in Section 3.1. In the end, the results suggest that position accuracy of independent SfM products without ground control were within 3–5 centimeters in the horizontal directions, and that the use of coincidental geographic points—without accurate survey position data—could be used to correct these offsets and produce sub-centimeter comparative DSM and orthoimage products (Figure 10d).

**Table 3.** Differences in 3D positions of distinctive geographical points in two independently processed Structure from Motion (SfM) products from the two survey days within the area mapped in Figure 10.

| Variable | Difference in Easting (m) | Difference in Northing (m) | Difference in Total Horizontal (m) | Difference in Elevation (m) |
|---|---|---|---|---|
| Mean | 0.031 | 0.035 | 0.047 | 0.001 |
| Median | 0.031 | 0.035 | 0.047 | −0.004 |
| St. Dev. | 0.003 | 0.002 | 0.001 | 0.010 |
| Count | 8 | 8 | 8 | 8 |

### 3.3. Field Operation Parameters

3.3.1. Number of Cameras

The number of cameras used in the SfM processing had a strong effect on the resulting data coverage, as shown with DSMs from different camera setups (Figure 11). For example, the use of a single downward-directed camera resulted in discontinuous coverage and about 47% less data than the four-camera setup (Figure 11a). Gaps in the single-camera DSM were largest and most extensive in the shallowest portions of the study area, where raw imagery suggests that image overlap in the along-track and across-track directions were compromised. This result is consistent with an overall narrower field of view for one downward-directed camera compared to multiple outward-directed cameras, which limited image overlap. In addition, our downward-directed camera had a longer lens (8 versus 6 mm), which further reduced its field of view, thereby resulting in SfM products that were generally unusable if only this camera was used (Figure 11a).

Mapping coverage substantially improved for two- and three-camera setups versus a single camera (Figure 11). These improvements can be attributed to a broader coverage, or "footprint," of the lakebed from each pass of the SQUID-5 system as a result of the placement and orientation of multiple cameras. For example, two or three cameras resulted in approximately 80% to over 90% of the study area lakebed coverage that occurred for the four-camera collection (Figure 11b–d). However, although the total along-track width (or "swath") for the imagery did not change for our two- and three-camera set-ups—all use the outside lateral cameras, which provide this broad swath—it is important to highlight that adding the third cameras significantly increased coverage and reduced the number and size of data gaps (Figure 11). Additionally, the data suggest that more coverage was obtained when the third camera was oriented outward (toward the forward direction) than straight downward. This is consistent with the outward-looking camera collecting somewhat more unique information within the complex lakebed landforms, in opposition of the more redundant information collected from the downward camera. That noted, our four-camera results, which included both the forward-directed and downward-directed cameras, provided even more coverage than either of the three-camera setups, and it also had a reduction in the number and size of data gaps (Figure 11). Interestingly, the added value of the third camera (~10 million more pixels) was roughly the same as the added value of the fourth camera (Figure 11).

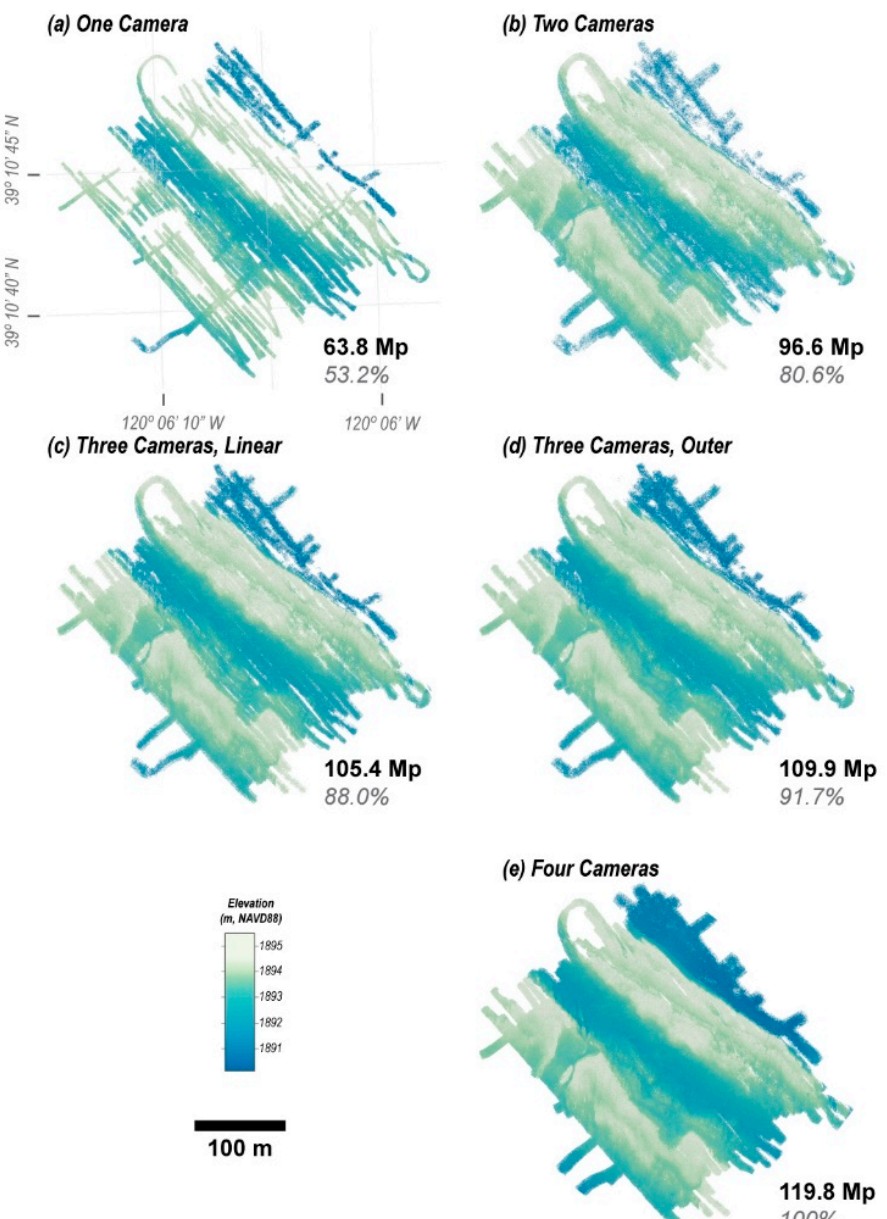

**Figure 11.** Digital Surface Model (DSM) output for the study area as mapped with subsets of the available cameras on SQUID-5. Each output provides the spatial extent of the DSM map and the total number of 25 mm x 25 mm pixels with elevation data (provided in millions of pixels, or Mp) and the ratio of each pixel count to the total pixels generated by the four-camera setup (shown in %).

### 3.3.2. Line Spacing

The presence of gaps in the final DSMs provided an opportunity to assess the functional along-track width of data collection of the SQUID-5 system, and hence, provide additional guidance about operational line spacing for surveys. For this, we found the initiation points of a total of 21 gaps in the DSM data derived from all four functional cameras. These gaps were chosen, as best as possible, where two adjacent survey pathways were straight, but slightly divergent, so that line spacing at the gap could be calculated. The data obtained from these initiation points shows that water depth has a first-order effect on the line spacing distance where gaps form (blue symbols; Figure 12). In fact, simple scaling of the field of view and geometry of the SQUID-5 system [12] suggests that complete coverage of flat lakebed without sensor tilting would occur at 2.17 times the water depth plus 0.93 meters (blue line; Figure 12), and this relationship closely mimics the

least-squares fitted line through these data (dashed blue line; Figure 12). However, several data gaps were found to occur at line spacing distances less than the theoretical values provided by the simple linear equation. Examination of these areas suggest that they were likely influenced by increased roughness of the lakebed, for example, from large cobble and boulders or sloped topography, which effectively resulted in a reduction of the surface area of the bed that could be imaged.

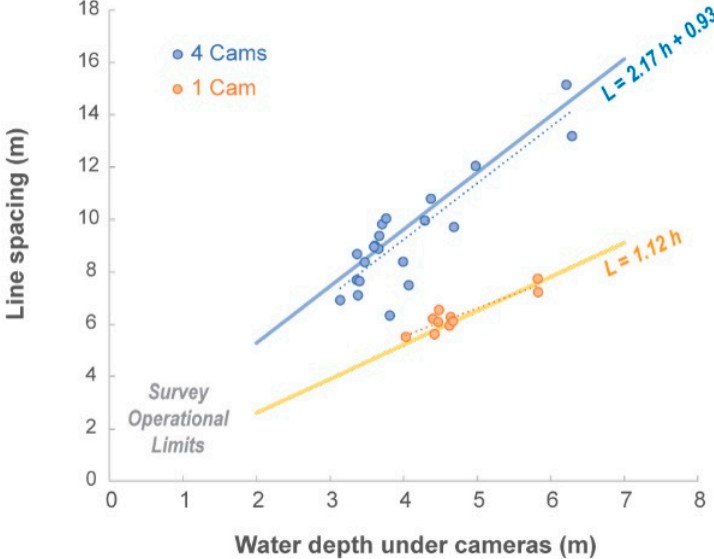

**Figure 12.** Comparison of the water depth and line spacing for locations where data gaps form in the Digital Surface Model (DSM) output for both a four-camera (blue symbols) and a one-camera analysis. The theoretical line spacings required for complete mapping assuming a flat bed, and the geometry and camera fields-of-views are shown with thick lines. The least-squared linear fits through the data are shown with dashed lines.

We also measured an additional 10 initiation points for data gaps in the single-camera SfM output. Gaps in this data product occurred at line spacings much smaller than the four-camera products, and the simple geometric scaling using just the field of view of our camera closely matches these data (orange symbols and lines; Figure 12). As such, it appears that simple field-of-view calculations from the geometry of the camera systems can be useful tools. However, these equations neglect the impacts of complex roughness elements of the bed and the operational challenges of achieving desired line spacings in field conditions that include winds, currents, waves, and other factors.

## 4. Discussion and Conclusions

The work described here shows that a synced GNSS–underwater camera system, such as the SQUID-5, can be successfully used to map areas large enough to be of significance to coral reef science with both high resolution and geospatial accuracy. Our two days of field work resulted in almost complete coverage of over 70,000 m$^2$ of shallow reef at 2.5 cm DSM resolution and 0.5 cm orthoimagery resolution. This operation was conducted in a lake with clear water conditions, which allowed for excellent opportunities for imaging the bed. Additionally, the weather conditions, which peaked with 0.5 m wind waves, did not negatively influence our ability to collect data. Substantial differences were observed in lighting conditions during the cruise, however, and these effects were normalized with the color correction technique combined with ortho-average SfM data processing. The latter element also resulted in reductions of the negative effects of underwater light caustics.

This survey operation benefited from several technological advances that allowed for the synchronous acquisition of imagery from multiple cameras and from a survey-grade GNSS system and the rapid storage and backup of the data captured. The operation

also benefited from intensive planning to evaluate the best camera settings (Table 1), theoretical line spacing, and real-time survey navigation communications to the boat's pilot. Additionally, the use and analysis of survey data while in the field allowed for near-real-time assessments of survey coverage and potential gaps. The inclusion of SfM basic image alignment data processing during non-survey hours, which are typically the nights between survey days, can allow for more thorough analyses of coverage while in the field (Figure 3). Although surveys could be conducted without some of these elements, they would likely be hindered by slower data collection speeds or longer survey operations to ensure complete coverage.

The most important technological element to the success of this surveying technique is the tightly coupled camera–GNSS data collection. Our uncertainty analyses suggest that the SfM positional errors scaled with the position accuracy of the image data, consistent with the findings of Hatcher et al. [12]. Therefore, offsets in the data collection timing between the camera and GNSS, or poor GNSS data, can introduce substantially larger errors than shown here. That noted, the multiple-centimeter uncertainty in DSM and orthoimagery data that was reduced to sub-centimeter with unsurveyed points was derived from only a small portion of the study area (~0.35%). A valuable next step in this work will be to expand the size of repeat mapping and change analyses to test whether metrics for repeatability are sustained over reef-scale survey areas (O(100,000 m$^2$)), and also whether offsets are constant over these larger spatial scales. There is the potential that 4D SfM data processing techniques, which utilize co-alignment analyses of multiple survey dates, may benefit in change detection over these reef-scale surveys [19].

The comparison of SfM products derived from different numbers of cameras provides important findings about collecting complete SfM data sets over complex reef settings. First, the theoretical swath widths that include the field-of-view geometry for different camera setups are generally useful for assessing survey line spacing. However, these calculations need to include the realistic assessments of the ship's ability to achieve line spacings in the field, the effect of reef complexity on the system's swath width, and additional goals for ground sample resolution and/or photo overlap. We found that our multiple-camera operations, especially those with three or four cameras, achieved fairly complete data coverage over these theoretical widths (Figure 11), suggesting that substantial areas of across-track overlap ("sidelap") are not needed. We hypothesize that the multiple-camera operations enhance overall alignment because of the camera-to-camera alignment that will occur with each triplet or quartet of imagery, which is something that could be evaluated in much more detail with further testing.

The presence of data gaps in our survey and their relationship with survey line spacing (Figure 12) provides valuable information for future operational planning. One example is that the line spacing for single-camera operations will need to be much smaller than multi-camera operations (Figure 12). Additionally, including outward-looking camera orientations for multiple-camera systems provides a means to capture imagery on different sides and aspects of complex reef shapes, which increases total coverage in SfM products (Figure 11). So, whereas single-camera operations are possible with underwater SfM [2–4,6,8], incomplete data collection from these systems can occur where the look angle of the camera does not adequately image complex features of the bed. Unfortunately, our fifth and backward-viewing camera was not operational during this survey, so we cannot evaluate whether additional camera perspectives to those shown here would have enhanced coverage of the study area.

The use of a longer lens in the central camera of SQUID-5 provided the more highly resolved imagery of the bed conditions compared to the outward-looking cameras (data can be observed in Hatcher et al. [18]). This increased resolution allowed for better derivation of orthoimagery products, especially directly underneath the SQUID-5 system. Additionally, this downward-looking camera was observed to increase overall mapping coverage, even though it provided a largely redundant coverage area (Figure 11).

In summary, we have shown that a coupled underwater camera and GNSS system can be used to effectively map complex benthic settings at resolutions and accuracies that should help with describing benthic conditions and ecosystems and measuring benthic change. These results can be realized without the use of surveyed ground control, but the inclusion of steady reference points can improve the accuracy of survey-to-survey change measurements. Because the data collected with this system are highly resolved and spatially accurate, they can also be used to calibrate and assist with the development of other state-of-the-art bathymetric mapping techniques, including the use of digital imagery from areal platforms that account and adjust for the complex refraction at the air–water interface [11,13] and lidar. Although the SfM techniques shown here are limited to settings with adequate water quality to photograph the bed from a floating platform, the resulting data not only provide highly accurate and highly resolved three-dimensional point clouds, but also color information for each mapped point of the study area. The combination of color—which can be corrected for water attenuation effects, as shown above—and the three-dimensional shape of the bed should provide ample opportunities to classify lakebeds and seafloors using techniques such as machine learning, and by doing so, improve assessments of the ecological and geological characteristics of benthic settings. As such, there is a great opportunity to map reef-scale characteristics and changes over time with the techniques presented here.

**Author Contributions:** G.A.H. led the engineering, bench testing, and fabrication of SQUID-5 hardware, electronics, and control software; led the fieldwork; and compiled co-author input. J.A.W. wrote the color correction software; composed the structure from motion data processing; and generated the resulting data products, validation, and analysis methods. C.J.K. performed the GNSS data analysis and application of the PPK corrections. A.C.R. provided guidance for SfM analysis, presentation, and interpretation. All authors have read and agreed to the published version of the manuscript.

**Funding:** Funding was provided by the following U.S. Geological Survey sources: (1) The Coastal and Marine Hazards and Resources Program through the Remote Sensing Coastal Change project; (2) The Coastal and Marine Hazards and Resources Program through the Marine and Field Operations project.

**Data Availability Statement:** The data sets generated for this study can be found in the publicly accessible repositories https://doi.org/10.5066/P9V44ZYS and https://doi.org/10.5066/P9934I6U (accessed on 1 April 2023).

**Acknowledgments:** Any use of trade, firm, or product names is for descriptive purposes only and does not imply endorsement by the U.S. Government. We would like to thank the other member of the USGS fieldwork team, Pete Dal Ferro, for his help with the chilly winter Lake Tahoe fieldwork, and the UC Davis Tahoe Environmental Research Consortium, Captain Brant Allen, and the crew of the R/V John Le Conte for their expert boat handling and generous sharing of their local knowledge.

**Conflicts of Interest:** The authors declare no conflict of interest.

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
