# Peer review of "Accurate Maps of Reef-Scale Bathymetry with Synchronized Underwater Cameras and GNSS"

_remotesensing, doi:10.3390/rs15153727_

Round 1
Reviewer 1 Report
The manuscript is well-written and organized, with clear objectives, methods, results, and conclusions. The authors have effectively communicated their research findings and provided adequate references to support their claims. After thorough evaluation, I am pleased to recommend the acceptance of this manuscript for publication in Remote Sensing.
Author Response
Dear Reviewer#1,
We sincerely thank you for your kind words and generously giving your time to review our manuscript.
Reviewer 2 Report
Dear authors,
Bathymetry has been an interesting topic for decades and thanks to recent developments and committed input by researchers like you to extend the scale and scope of research to new fronters. This article is an interesting read and, on my side, here are the recommendations and suggestions to better the quality and content of the manuscript. I will respond by each section and whereas in some instances, I will use the page and line number. My comments as follows.
1. Introduction
Line 32-54: I believe, the background research review is shallow and need to be extended. I strongly recommend authors to somewhat introduce the main topic i.e., bathymetry and why it is important, and where we stand now. In this regard, LiDAR should be considered and mentioned as an active sensor to measure shallow water bathymetry. In comparison, why do we need SfM technology when LiDAR seems to work fine or where it doesn't work and SfM is a better alternative? That would establish and good research review for readers with little or no prior knowledge of bathymetry.
Line 55-80: That should include related work section and better to make two sub-sections. Section 1, documenting a bit history of development in systems used and some fundamental details of the systems deployed in this study. I think details related to Lake Tahoe is bit earlier to mention here in the introduction part. That could be move to Section 2: Contributions part where authors can mention their research objectives and areal extent they aim to cover. Figure 2 therefore should move to study area in later part of the paper where authors can highlight significance of selected sites and system deployment.
Line: 66-67 instead of 1000s and 100s use the standard notation i.e., m square. This one sound less sound and (s) is standard unit for time -second. Stating area in square meters make more sense than length. Please adhere these standards throughout your manuscript.
Following the above format, next section should be study area and system used.
Figure 1. label each plot with (a), (b), (c), and (d) format with captions. In the following manners, b and d are same thing therefore remove one of them probably d. If possible, add a scale in the image so that user can get the idea of the size of the system. Enlarge the size of the Fig. to fit in whole page space. In present, it is hard to comprehend the Fig.
Authors have designed the section 2.2. study Site, Therefore, I highly recommend moving all related text and figures under these sections as the authors mentioned study are earlier e.g., Figure 2 and related content should go under section 2.2. Move section 2.2. earlier than Methodology. Please use a standard template or you can consider the following paper. Maybe a graphical depiction of how the system works probably carries more information than an actual image of the system. Please check this.
Blockchain for unmanned underwater drones: Research issues, challenges, trends and future directions - ScienceDirect
Line 81-92: This could be another sub-heading with the title "Contributions".
Methodology
Instrumentation. It would be a much better approach to detail system description in this section including a high-resolution system image.
Line: 103-106: FLIR machine vision cameras: define the acronym (FLIR). Please the camera specifications better use a table and enlist all particular details of the camera model as this become non-trivial in later sections. Spectral bands used by the FLIR system include wavelengths, this is important to know as sunlight travels through the water column. The time of the day is also important to consider as sunlight angle of incidence changes with time. Does the camera calibration was considered?
Line 197: I think it should be done manually if I am correct.
Line: 98-109 warrants a graphical depiction to demonstrate how it works with the overall layout of the system over the water surface. The above-provided article is a guiding principle.
Line 125 - 142: 5 units 5 MP cameras resulting in a Terabyte scale of data is a somewhat overwhelming statement. In comparison, modern UASs equipped with 45 MP camera results in data in GB, not TB how come your systems with comparatively lower resolutions i.e., 5 MP can result in datasets of size in TB? Please justify your data acquisition and storage approach and also the enormous size of the dataset captured by SQUID-5.
Move section 2.2. earlier in the text.
2.2. Study Site
sufficient water clarity. Does not sound like a standard scientific notation. water clarity is measured in Secchi Depth and therefore sets a standard protocol to collect and deploy water-borne imaging sensors. The visual water clarity might be different for different observers. During the survey, I hope the authors took some photos of the site environment from observers' perspectives. Please consider that as well. The verbal description should be supported by some visuals (Line 146-163).
Figure 2. What are the yellow lines and what are the green lines in Fig. Please provide a legend.
2.3. Field Operations
The authors did not document what was the percentage of forward and lateral overlaps between images collected by the system. Field operation and field data collection sound the same thing so merge into a single section.
Fig. 3 and 4. Enlarge text font size.
Describe methods used in ref. [19] and [21]. It is important to consider these details.
Line 225-229. How did the reconciliation happen? What software applications/tools/codes/ were used or developed?
Line 260: I think the author means Figure. 5, please check.
Fig. 5. Two of three examples with unique information are enough. The authors lack the details of examples in the main body text. Also, the authors need to provide an inset map of the study area from which the sample was taken. The same is true for Fig.6.
Figure 7. Show relative elevation setting up the lowest values to zero so that the depth can make more sense. Also, enlarge the text and font size.
The same is true for the remaining figures to improve their readability.
Now recommendations. Why did authors not use LiDAR bathymetry or any other bathymetry product for accuracy evaluations?
The quality of the present lack several improvements including adding/revising several sections and subsections. I recommend authors find similar articles and use a published paper as a template to organize and present their results. For instance, authors are not consistent in their approach, they can select a small area out of the entire study site to present consistent results as shown in Fig 5. Figure 6, Fig 8.
Also, I do not see authors' results and data from the survey 2020? why not include or authors can leave this section.
spell checks and typos are detected.
Author Response
Dear Reviewer#2,
We sincerely thank you for your feedback and generously giving your time to review our manuscript. Individual items are addressed below.
- Introduction
Line 32-54: I believe, the background research review is shallow and need to be extended. I strongly recommend authors to somewhat introduce the main topic i.e., bathymetry and why it is important, and where we stand now. In this regard, LiDAR should be considered and mentioned as an active sensor to measure shallow water bathymetry. In comparison, why do we need SfM technology when LiDAR seems to work fine or where it doesn't work and SfM is a better alternative? That would establish and good research review for readers with little or no prior knowledge of bathymetry.
Although having a much more in-depth research review on bathymetric mapping could be a resource for the research community (especially those with little experience as the reviewer notes), we see that this would be outside the focus of the paper and would necessitate a massive increase in the paper length. Not only would we have to include LIDAR in this review, but several other sound-based mapping techniques (single-beam, multi-beam, swath sonar, and their waveform interpretations) would have to be compared and contrasted. This would be a great addition for a stand-alone review paper on bathymetric mapping, but it is much too complex for this paper.
Line 55-80: That should include related work section and better to make two sub-sections. Section 1, documenting a bit history of development in systems used and some fundamental details of the systems deployed in this study. I think details related to Lake Tahoe is bit earlier to mention here in the introduction part. That could be move to Section 2: Contributions part where authors can mention their research objectives and areal extent they aim to cover. Figure 2 therefore should move to study area in later part of the paper where authors can highlight significance of selected sites and system deployment.
We covered the fundamental details and development of this system in our first paper (Hatcher et al., 2020). Repeating those details here would be redundant, so we refer the reader to the original paper. Minor changes have been made to highlight these matters.
Secondly, we have used a very common style of providing a brief description of the study site in the Intro. This helps the reader know – in a very broad, introductory sense – what the work in the paper is. Thus, no changes made.
Line: 66-67 instead of 1000s and 100s use the standard notation i.e., m square. This one sound less sound and (s) is standard unit for time -second. Stating area in square meters make more sense than length. Please adhere these standards throughout your manuscript.
Following the above format, next section should be study area and system used.
Both figures of length and area are important. Length is especially important in mapping coral reefs, which can be highly elongated in the longshore direction. Thus, we include reference to both units.
Figure 1. label each plot with (a), (b), (c), and (d) format with captions. In the following manners, b and d are same thing therefore remove one of them probably d. If possible, add a scale in the image so that user can get the idea of the size of the system. Enlarge the size of the Fig. to fit in whole page space. In present, it is hard to comprehend the Fig.
Thank you. We’ve added labels and scalebar as suggested. However, we choose to include both panels 1b and 1d. 1b which indicate how the vehicle floats on the water surface and 1d which is now referenced later in the document when describing the winter working conditions at the field site.
Authors have designed the section 2.2. study Site, Therefore, I highly recommend moving all related text and figures under these sections as the authors mentioned study are earlier e.g., Figure 2 and related content should go under section 2.2. Move section 2.2. earlier than
This is not a typical formatting style. Authors commonly include references to study area features in other sections than the Study Area. For example, the Introduction section commonly ‘introduces’ the study area and references the site map. Additionally, there may be references to the study area in the Methods section, where different methods were used in different areas. Thus, we do not choose an overly rigid and prescriptive organizational structure.
Methodology. Please use a standard template or you can consider the following paper. Maybe a graphical depiction of how the system works probably carries more information than an actual image of the system. Please check this. [Blockchain for unmanned underwater drones: Research issues, challenges, trends and future directions – ScienceDirect]
Thank you for the suggestion, we are using the standard template provided by MDPI. A detailed graphical depiction and description are provided in our first paper describing the engineering of the system (Hatcher et al., 2020) and referred to throughout this paper.
Line 81-92: This could be another sub-heading with the title "Contributions".
Thank you for this comment. We do not see that an additional sub-heading would add value to the organizational structure.
Methodology
Instrumentation. It would be a much better approach to detail system description in this section including a high-resolution system image.
As noted above, the system is described in full detail in (Hatcher et al., 2020), which is cited throughout this paper.
Line: 103-106: FLIR machine vision cameras: define the acronym (FLIR). Please the camera specifications better use a table and enlist all particular details of the camera model as this become non-trivial in later sections. Spectral bands used by the FLIR system include wavelengths, this is important to know as sunlight travels through the water column. The time of the day is also important to consider as sunlight angle of incidence changes with time. Does the camera calibration was considered?
Teledyne FLIR is the brand name of the camera manufacturer; we’ve corrected this in the manuscript. It’s an RGB wavelength camera and specific camera detail has been added to the manuscript as suggested.
Line 197: I think it should be done manually if I am correct.
Actually, “minimally” is our intended word. The initial processing done in the field was for on-site rapid quality assurance. The final processing was done after the survey was complete and the GNSS base station ephemeris data were available.
Line: 98-109 warrants a graphical depiction to demonstrate how it works with the overall layout of the system over the water surface. The above-provided article is a guiding principle.
Please note comments above about this topic. (Hatcher et al., 2020) and its supplementals cover these details.
Line 125 - 142: 5 units 5 MP cameras resulting in a Terabyte scale of data is a somewhat overwhelming statement. In comparison, modern UASs equipped with 45 MP camera results in data in GB, not TB how come your systems with comparatively lower resolutions i.e., 5 MP can result in datasets of size in TB? Please justify your data acquisition and storage approach and also the enormous size of the dataset captured by SQUID-5.
We have provided a better description of the data collection requirements for this system, which are unique owing to the low ‘flight’ altitude from the lakebed.
Move section 2.2. earlier in the text.
No. Sorry. It fits well here.
2.2. Study Site
sufficient water clarity. Does not sound like a standard scientific notation. water clarity is measured in Secchi Depth and therefore sets a standard protocol to collect and deploy water-borne imaging sensors. The visual water clarity might be different for different observers. During the survey, I hope the authors took some photos of the site environment from observers' perspectives. Please consider that as well. The verbal description should be supported by some visuals (Line 146-163).
The sentence fragment you mention is only part of a further qualified sentence: “sufficient water clarity that the bottom can be imaged from just below the surface using natural sunlight in depths as great as 8 meters.” It is just one item of a qualitative list of desirable characteristics considered during our fieldwork site selection and not intended to be a quantitative parameter.
All "raw" imagery used to generate data products described in this manuscript are available on our referenced data release website and available for the reader for download and viewing.
Figure 2. What are the yellow lines and what are the green lines in Fig. Please provide a legend.
The colors indicate the tracks of the survey vehicle during two separate days of operations. Thank you for the suggestion the figure has been updated with a legend.
2.3. Field Operations
The authors did not document what was the percentage of forward and lateral overlaps between images collected by the system. Field operation and field data collection sound the same thing so merge into a single section.
We agree that the along track and lateral image overlap should be reported and have added this information to the manuscript.
Fig. 3 and 4. Enlarge text font size.
We agree that the text very small as submitted. However, the full resolution figures are embedded within our submission document. We anticipate that the journal typesetters will size the figures as required to meet their standards and typesetting style.
Describe methods used in ref. [19] and [21]. It is important to consider these details.
We provide a multitude of the details of these techniques in the paper, so the work is completely reproducible.
Line 225-229. How did the reconciliation happen? What software applications/tools/codes/ were used or developed?
The reconciliation was done by simply matching the timestamp used to generate the image filename with the timestamp saved in the GNSS event records using Microsoft Excel. We have further indicated that in the manuscript.
Line 260: I think the author means Figure. 5, please check.
We did intend this to reference figure 4. This point in the manuscript is describing the workflow which is illustrated in the flowchart of figure 4. Figure 5 illustrates the resulting products which are discussed next in the manuscript.
Fig. 5. Two of three examples with unique information are enough. The authors lack the details of examples in the main body text. Also, the authors need to provide an inset map of the study area from which the sample was taken. The same is true for Fig.6.
The large photo set provides a great diversity of settings and color correction results. Readers should generally respond well to a series of examples. There are no prescribed rules that every figure subplot needs to be detailed in the text.
Figure 7. Show relative elevation setting up the lowest values to zero so that the depth can make more sense. Also, enlarge the text and font size.
The water depth is constantly fluctuating in lakes, so the standard procedure is to report lake bottom elevation relative to a local vertical datum. The approximate average water surface elevation at the time of the survey was included in the figure caption.
We agree that some of the figure text very small as submitted. However, the full resolution figures are embedded within our submission document. We anticipate that the journal typesetters will size the figures as required to meet their standards and typesetting style.
The same is true for the remaining figures to improve their readability.
We agree that some of the figure text very small as submitted. However, the full resolution figures are embedded within our submission document. We anticipate that the journal typesetters will size the figures as required to meet their standards and typesetting style.
Now recommendations. Why did authors not use LiDAR bathymetry or any other bathymetry product for accuracy evaluations?
The orders of magnitude difference in spatial resolution make comparison of LiDAR and the SfM products generated from our technique problematic. Absolute geospatial system accuracy was evaluated more thoroughly in our earlier paper (Hatcher et al., 2020). The goal of this work was to evaluate its capacity to collect an internally consistent data over a larger area and from separate data collection days.
The quality of the present lack several improvements including adding/revising several sections and subsections. I recommend authors find similar articles and use a published paper as a template to organize and present their results. For instance, authors are not consistent in their approach, they can select a small area out of the entire study site to present consistent results as shown in Fig 5. Figure 6, Fig 8.
The areas illustrated in Figures 5,6, & 8 were chosen specifically to highlight the properties of the data products or processing results and or artifacts where they were most obvious and easy to clearly describe in the associated manuscript text.
Also, I do not see authors' results and data from the survey 2020? why not include or authors can leave this section.
As described in the manuscript text the GNSS data were compromised to such a degree that SfM data products could not be generated which met USGS data publication standards and therefore were not released. However, the initial survey was used to verify our sampling technique and develop initial baseline camera settings, also indicated in the manuscript.
Reviewer 3 Report
This paper presents a novel method for mapping shallow reefs at high accuracy and resolution. In general, this work is interesting and has some potential. Some minor revisions should be made before publication.
1) Please highlight the contributions of the work before publication.
2) Figure 3 covers the GNSS data processing. Especially, the GNSS data are always polluted by bias and outliers. Some related work has proposed efficient algorithms to alleviate the above issue: autonomous vehicle kinematics and dynamics synthesis for sideslip angle estimation based on consensus kalman filter, automated vehicle sideslip angle estimation considering signal measurement characteristic. Thus, the above work should be included in the reference and discussed to help understand the paper.
3) For the point cloud classification, currently, there are some deep learning-based methods to enhance the performance: an automated driving systems data acquisition and analytics platform, hydro-3d: hybrid object detection and tracking for cooperative perception using 3d lidar. The above classical methods should be included to ensure the integrity of relevant work research.
4) The work limitations and future work should be highlighted at the end of the paper.
5) Please provide the parameter index of the sensor as it plays an important role for the accuracy.
Author Response
Dear Reviewer#3,
We sincerely thank you for your feedback and generously giving your time to review our manuscript. Individual items are addressed below.
1) Please highlight the contributions of the work before publication.
We have highlighted the contributions of this work in both the abstract and conclusion. It is difficult to address this suggestion because it does not indicate specific shortcomings.
2) Figure 3 covers the GNSS data processing. Especially, the GNSS data are always polluted by bias and outliers. Some related work has proposed efficient algorithms to alleviate the above issue: autonomous vehicle kinematics and dynamics synthesis for sideslip angle estimation based on consensus kalman filter, automated vehicle sideslip angle estimation considering signal measurement characteristic. Thus, the above work should be included in the reference and discussed to help understand the paper.
It should be noted that the SQUID-5 is not an autonomous vehicle, but rather a towed sensor, with no IMU. Were it so, vehicle sideslip angle (VSA) estimation could indeed be a valuable parameter to feed back into the control system. Instead, we take advantage of the benefits of post-processing the GNSS data after the precise ephemeris and clock data are available, which allows the GNSS processing software (NovAtel’s GrafNav) to robustly filter and correct outliers. GrafNav makes extensive use of Kalman filtering in its processing engine. The SfM software (Agisoft Metashape) then minimizes any residual error remaining after the GNSS processing and determines the orientation of each camera via bundle adjustment and multi-view stereo techniques.
3) For the point cloud classification, currently, there are some deep learning-based methods to enhance the performance: an automated driving systems data acquisition and analytics platform, hydro-3d: hybrid object detection and tracking for cooperative perception using 3d lidar. The above classical methods should be included to ensure the integrity of relevant work research.
We appreciate this note on new techniques for point cloud classification. We certainly look forward to investigating these techniques for future work. However, classification is not part of this study, our goals were to evaluate how well our system could generate larger area mapping products on a scale useful to reef science. Now that we have know we can achieve these goals we will be looking forward to best techniques to classify RGB point clouds in future work.
4) The work limitations and future work should be highlighted at the end of the paper.
Thank you for the suggestion. We’ve added some material to section 4. Discussion and Conclusion.
5) Please provide the parameter index of the sensor as it plays an important role for the accuracy.
This omission has also been flagged by other reviewers. The useful range of camera parameter settings which were derived during the fieldwork have been added to the manuscript in Table 1.
Round 2
Reviewer 2 Report
The manuscript has been improved. I am satisfied with author's feedback and revisions.